# Yoga and Qigong for Health: Two Sides of the Same Coin?

**DOI:** 10.3390/bs12070222

**Published:** 2022-07-03

**Authors:** Paula Boaventura, Sónia Jaconiano, Filipa Ribeiro

**Affiliations:** 1IPATIMUP—Institute of Molecular Pathology and Immunology of the University of Porto, 4200-135 Porto, Portugal; filipa.ribeiro@gmail.com; 2i3S—Instituto de Investigação e Inovação em Saúde, Universidade do Porto, Rua Alfredo Allen 208, 4200-135 Porto, Portugal; 3FMUP—Faculty of Medicine, University of Porto, 4200-319 Porto, Portugal; 4EAAD—School of Architecture Art and Design, University of Minho, 4800-058 Guimarães, Portugal; id5928@alunos.uminho.pt

**Keywords:** yoga, qigong, mind–body therapies, health benefits, breathing, movement

## Abstract

Yoga and qigong are ancient mind–body practices used in the East for thousands of years to promote inner peace and mental clarity. Both share breathing techniques and slow movements and are being used as alternative/complementary approaches to the management of disease, especially chronic problems with no effective conventional treatments. However, information comparing the health benefits of both approaches is scarce, and the choice between yoga or qigong practice may only depend on patients’ preferences or practice availability. The aim of the present paper was to overview yoga and qigong use for health benefits under different pathological conditions. Yoga and qigong seem to have similar effects, which might be expected, since both are similar mind–body approaches with the same concept of vital life-force energy and the practice of meditative movements. Problematic research issues within the literature on yoga and qigong are the small sample sizes, use of different styles, significant variance in practice duration and frequency, short duration of intervention effects, and the usage of a non-active control group, thus emphasizing the need for further high-quality randomized trials. Studies comparing yoga and qigong are warranted in order to assess differences/similarities between the two approaches for health benefits.

## 1. Introduction

Yoga and qigong are ancient mind–body practices used in the East for thousands of years to promote inner peace and mental clarity. Both share breathing techniques and slow movements, which can be easily learned by anyone, empowering the practitioner with a tool for psycho-spiritual growth. Yoga and qigong are based in the same concept of a vital life-force energy that sustains life everywhere, which in yoga is called *prana* and in qigong is called *qi*. They place emphasis on attending to interoceptive, proprioceptive, and kinesthetic qualities of experience [1].

Both yoga and qigong emphasize three common components in their fundamental practices: (1) stretching of muscles, tendons, and ligaments, where thousands of proprioceptive receptors are located; (2) controlled breathing leading to the harmonization of the somatic and autonomic nervous systems; (3) the obtention of a state of tranquility of the mind, which can be considered as meditation [2].

Qigong is a Chinese traditional medicine that uses vegetative biofeedback therapy to promote health and wellbeing and to treat medical conditions [3]. It combines gentle body movements with breathing and mindfulness [4,5]. Traditionally, qigong has been defined as the harmonization of *qi* (the internal vital energy of the body) and blood in the body, aiming to prevent disease and improve health [6,7]. The biophysical effects of qigong as a vegetative biofeedback therapy can be measured and quantified using various methods, namely, the measurement of the electrical potential of the skin [3]. Qigong is particularly appropriate for older people due to its gentle and smooth movements [8]; qigong movements are usually slower and gentler than yoga movements.

While yoga has its roots in Indian Vedic scriptures, qigong emerged from Chinese Taoism; however, they may be considered as different paths to the same goal, given that both aim to improve body health, to quiet and clarify the mind, and to strengthen connection to the inner soul and humanity [9]. Historically, yoga and qigong have different movements, postures, and focuses, but they both similarly use the breath to move energy and invoke a meditative state. Their overall purpose is the same, even if the way in which they achieve it is a little different. Yoga began as more of a spiritual practice, while qigong emerged as a practice for health preservation and is associated with martial arts. In its spiritual approach, yoga’s postures were originally created for building muscles so that the practitioner would be able to perform seated meditation for hours. Qigong, on the other hand, has less of a muscular focus, using more flowing movements, which are physically easier to practice. One final difference is that qigong (once one progresses past a beginner level) mostly focuses on balance, while a typical yoga session will probably only include one or two balancing poses.

Currently, in Western culture, yoga is considered a complex system of postural exercises combined with breathing, concentration, and meditation techniques [4]. Yoga, as a mind–body strategy, has the largest body of evidence in favor of potential health benefits, either due to the greater volume of studies [10,11] or to the fact that it is the mind–body practice that is most commonly used in the West [11]. Qigong is less known outside China [2] compared to yoga. In fact, searching on PubMed for the word “yoga”, 6132 articles were found, compared with 880 for the word “qigong”. Using both words together, only 146 articles were found. Qigong research publications have been gradually increasing, but reports on study types, participants, qigong interventions, and outcomes are diverse and inconsistent [12]. This emphasizes the need for trials of a high methodological quality and with sufficient sample sizes to verify the effects of qigong in health and disease management [12].

Most yoga studies only evaluated the exercise effects of yoga, although yoga is a practice that is not just limited to *asana* (postural exercises) or *pra**nayama* (breathing exercises) [13]. This occurs due to the ease of teaching and learning *asana* and *pra**nayama*, as well as their consequent assimilation into life style [13,14], but even *asana* is not a mere posture—it is the aligning of the body with complete involvement of the mind, consciousness, and intelligence [15]. *Asana* may protect against depressive symptoms, particularly when triggered by stress [16]. Each posture has a specific alignment, which influences the distribution of *prana* (commonly translated as vital energy) in the body, influencing its energetic state.

The aim of this paper is to overview the use of yoga and qigong for health benefits under different pathological conditions, in an effort to find out if yoga and qigong are used similarly or differently according to diverse pathologies and patients’ preferences. These mind–body therapies (MBTs) were chosen because both practices originally accompanied a form of Eastern medicine, and are presently used in the West as complementary therapeutic approaches for health benefits, without knowing which one is the best for the management of a certain pathology.

## 2. Methods

We performed a literature search in PubMed using the words “yoga”, “qigong”, or “yoga and qigong” and included studies published before May 2021. We focused our analysis on the “yoga and qigong” search, which retrieved 145 studies, since this was the focus of our paper. We excluded all case reports and mainly included systematic reviews and/or meta-analyses of randomized control trials (RCTs). Since it was not possible to approach all of the pathologies studied, we decided to include the ones that were more commonly considered in the literature reviewed. We also used articles addressing the characteristics of yoga and qigong users.

Relevant websites, including those of specialists in yoga and qigong, were also included in the search to obtain general information on yoga and qigong practices and for comparisons between the two techniques—information that could not be obtained from the papers retrieved from PubMed.

## 3. Yoga and Qigong Users

In Western culture, yoga is more popular than qigong, and instruction in it is more easily obtainable [17].

The number of yoga practitioners in the United States of America (USA) has considerably increased in recent years [18]. An increase in yoga, qigong, or tai chi practice was observed—from 5.8% in 2002 to 14.5% in 2017 [19]—but Lauche et al. reported a significant increase in yoga use (86%) compared to a slight increase in qigong use (about 4%) [20]. One possible explanation for this difference is that yoga has been publicly advertised in the press much more aggressively than qigong [20].

Gender differences in the practice of tai chi and qigong have been reported. Yoga classes are predominantly attended by females, but the reasons for this difference have not yet been well documented [19]. Yoga users are more likely to be white, female, young, and college educated, and benefits have been reported for musculoskeletal conditions and for mental health [21]. In Australia, although both yoga and qigong were found to be dominated by women, qigong appears to have a stronger appeal to men when compared to yoga [22]. In contrast, a yoga practitioner in India is more likely to be male, between 21 and 44 years of age, high-school educated, and a student [23].

In the USA and Australia, the most common reason to practice yoga is physical fitness, followed by disease management [23].

## 4. Yoga and Qigong in Health and Disease

MBTs, such as yoga and qigong, are currently used in the West as alternative/complementary approaches to the management of disease, especially chronic problems for which there is no effective conventional treatment. These two popular systems of self-performed bodily exercises are applied to the maintenance of a healthy state of the body and mind [2].

### 4.1. Immune System and Inflammation

Several conditions that are responsive to yoga and qigong practices, such as fatigue, depression, and pain, comprise inflammatory processes [24], which may explain researchers’ growing interest in the impact of MBTs on inflammatory markers [25,26]. Overall, the findings of Morgan et al. suggest that MBTs may reduce inflammation, particularly among clinical populations, as evidenced by the significant reductions in C-reactive protein (CRP) [27].

Vanketesh et al. [26] observed a downregulation of the inflammatory response in chronic disease through yoga practice. This had been previously reported by Falkenberg et al. [28], even though the existing evidence is not entirely consistent. In particular, decreases in IL-1 beta, IL-6, and TNF-alpha have been described in RCTs [28]. The authors hypothesized that longer periods of yoga practice are necessary in order to attain consistent effects on circulating inflammatory markers.

According to Bower and Irwin [25], the evidence for the effects of MBTs on IL-6 and other inflammatory markers was diverse, with the majority of the studies showing no changes. Of note, the studies with no significant alterations of the inflammatory markers presented beneficial effects on various symptoms, improving the patients’ health status [25,29].

### 4.2. Lower Back Pain

Lower back pain (LBP) is a condition affecting most people, and it results in functional limitations due to the lack of an effective treatment [30].

Yoga is a good therapeutic approach to LBP treatment, with the majority of the RCT studies showing pain reduction, improvement of psychological distress, and increased energy levels [30]. However, there is limited research on how the practice of yoga relieves back pain [30]. It is assumed that the mind–body connection in yoga, which is achieved by mentally focusing on breath and movement, may provide benefits to LBP patients [30]. Through this focus, yoga may modify the perception of pain; it has been shown that yoga practitioners have larger pain thresholds during thermal and pain threshold tasks [14]. For spine-related conditions, the core point is that yoga postures make use of unusual positioning in which the body weight is supported by the arms, hips, or muscles of the body. By pitting one muscle group against another, such as in forward bends, each group—agonist and antagonist—is equally exerted and, thereby, strengthened.

Concerning qigong, there are very few studies on chronic LBP, so it is not clear if qigong can be useful in LBP treatment [30]. It was observed that qigong significantly decreased pain intensity, back functional impairment, heart rate, and respiratory rate, and it increased range of motion, core muscle strength, and mental status in office workers with chronic non-specific lower back pain compared to a waiting-list control group [31]. On the other hand, Park et al. conducted a narrative review of RCTs that included an active control group, reporting that qigong was not more effective than physical exercise or other alternative therapies [30]. In the only study comparing the effects of yoga and qigong on the reduction of chronic LBP, Teut et al. [4] showed that neither yoga nor qigong was better than no treatment in reducing pain and increasing quality of life (QoL).

### 4.3. Mood Disorders, Sleep Disturbance, Cognitive Impairment, and QoL

Depression is excessively common in people with chronic pain; MBTs (including both yoga and qigong) provide small to moderate reductions of depressive symptoms in the chronic pain context [32]. However, this research area is still underdeveloped [32].

People come to yoga classes with a wide array of emotional conditions. For those experiencing depression, yoga can be helpful, as yoga classes may create space for healing emotional traumas and for achieving a steadier sense of serenity [33]. Yoga may serve as an efficacious supplement to pharmacotherapy, psychotherapy, and healthy lifestyle interventions for people with mental disorders [34].

Owing to the poor quality of most of the conducted research and to the diversity of the conditions approached, Kirkwood et al. [35] stated that, at that time (2005), it was not possible to affirm that yoga was effective for treating anxiety or anxiety disorders [35]. After this report, Cramer et al.,, in 2018, performed a meta-analysis of RCTs concluding that yoga might be beneficial for treatment of anxiety when compared to untreated controls or to an active control group [36]. In a more recent systematic review, James-Palmer et al. [37] showed that yoga may reduce anxiety and depression in young people regardless of health status and intervention characteristics. Of note, both authors emphasized that the studies analyzed presented a weak to moderate methodological quality [36,37].

Gallegos et al. [38] suggested that yoga is a promising complementary strategy for the treatment of post-traumatic stress disorder. Again, the authors stated that further research is needed, namely to comprehend how yoga complements traditional psychotherapy approaches or affects various aspects of disorders that are not addressed by these conventional therapies [38].

Stress reduction seems to be one of the most important benefits of qigong, in which the mind is used to guide activation and deactivation patterns through imagination [39]. According to Saed et al., qigong has shown inconsistent effectiveness as a complementary treatment for depression and anxiety [40]. On the other hand, according to Yeung et al. [41], there is preliminary evidence that qigong may be potentially beneficial for the management of depressive and anxiety symptoms in healthy adults and patients with chronic illnesses. Guo et al. found that qigong-based exercises may be effective for alleviating depression symptoms in individuals with major depressive disorder [42]. In the context of drug abuse, Liu et al. [43] suggested that qigong may alleviate anxiety symptoms. However, the authors recommended that their results should be interpreted with caution given the limited numbers of RCTs and their methodological weaknesses [41,42,43].

Zou et al. [44] conducted a meta-analysis of RCTs on meditative movements (specifically tai chi, qigong, and yoga), suggesting that they may have positive effects on the treatment of major depressive disorder, without causing important adverse events. On the other hand, Vancampfort et al. [34] recently stated in an umbrella review that the methodological quality and content strength of qigong trials for mental disorders are currently lacking.

In comparison with an inactive control, yoga and qigong significantly reduced insomnia symptoms and improved sleep quality [45]. These practices may also improve cognitive function [46,47] everyday activities functioning, memory, resilience, and mindfulness in older adults with mild cognitive impairment, but further research evidence is still needed to make a more conclusive statement [47]. Yoga may improve cognitive functions—particularly attention and verbal memory—in patients with mild cognitive impairment [48]. This may occur through improved sleep, mood, and neural connectivity [48]. Additionally, Weber et al. [49] found that yoga and qigong seem to positively influence QoL, depressive symptoms, fear of falling, and sleep quality in old adults without mental health conditions.

### 4.4. Cardiovascular, Neurologic, Respiratory, and Metabolic Diseases

The practice of qigong reduces the systolic and diastolic blood pressure in comparison with those of control groups [6,50,51]. Qigong may be an alternative non-pharmacological strategy for hypertension management, namely, in an elderly population group that usually takes too many medications [6]. In patients in an early stage of recovery from stroke, qigong showed more changes than conventional respiratory training in the improvement of trunk control ability, respiratory muscle functions, and ability to perform daily life tasks [52].

Yoga might be considered as an effective adjuvant for patients with various neurological disorders, such as stroke, Parkinson’s disease, multiple sclerosis, epilepsy, Alzheimer’s disease, dementia, headache, myelopathy, and neuropathies [53].

Reychler et al. [54] found that qigong and yoga improve the main symptoms of chronic obstructive pulmonary disease. Both therapies produced an effect that was slow but increased with time, with a good rate of adherence and long-lasting effects. Yoga (including *pranayama*) has been suggested as an adjuvant therapy in the treatment of childhood asthma, although it cannot yet be recommended as a standard of care due to the insufficiency of data on its efficacy [55]. A Cochrane review on this pathology showed moderate-quality evidence that yoga may provide small improvements in asthma patients’ symptomatology [56]. In patients with respiratory diseases, namely, asthma, *pranayama* has physiological and psychological benefits, although more high-quality RCTs are required to obtain definitive evidence [57].

Yoga may promote significant improvements in the management of type 2 diabetes (DM2), as shown by the glycemic control (including HbA1c) [58,59], lipid levels [58,59], and body composition [59]. More limited data suggest that yoga may also lower oxidative stress and blood pressure, enhance pulmonary and autonomic function, mood, sleep, and QoL, and reduce medication use in adults with DM2 [59]. Qigong was not found to have any advantages in reducing fasting blood glucose or postprandial blood glucose in patients with DM2, but demonstrated better control of HbA1c than that of other aerobic exercises [60].

Cramer et al. [61], who based their meta-analysis on the use of yoga for the management of metabolic syndrome, stated that no recommendation could be made for or against yoga’s effects on the parameters of metabolic syndrome. However, they also stated that, in spite of the methodological problems found in the evaluated studies, yoga could be preliminarily considered an effective intervention for reducing waist circumference and systolic blood pressure in people with metabolic syndrome who do not want to practice conventional forms of exercise [61]. Regarding qigong, Zou et al. [51] suggested that it might be an effective intervention for improving the risk factors for cardiovascular disease in the metabolic syndrome.

### 4.5. Cancer

Yoga and qigong have been shown to improve the anxiety and mood changes that are commonly associated with pain in cancer patients, even though these MBTs were not able to reduce the pain [62]. Evidence supports recommending yoga for the improvement of psychological outcomes and, probably, physical symptoms in adult cancer patients undergoing treatment [63]. For childhood patients, the evidence is insufficient [63].

Gentle hatha and restorative yoga are effective practices for treating sleep disruption, cancer-related fatigue, cognitive impairment, psychosocial distress [63,64], and musculoskeletal symptoms in cancer survivors and cancer patients under treatment [64].

Agarwal et al. [65] reported that most of the studies reviewed showed that yoga improved the physical and psychological symptoms, QoL, and markers of immunity of cancer patients, regardless of some methodological deficiencies. That said, their study supports the inclusion of yoga in conventional cancer care [65].

As an adjuvant therapy, yoga seems effective for QoL improvement in women with breast cancer, with better results with increasing intervention time [66]. Specifically, it seems to increase the psychological and social wellbeing of breast cancer survivors by helping to restore their body image and self-esteem and easing the return to their previous daily lives [66]. Future directions for yoga research in oncology should comprise: enrollment of participants with various cancer types, standardization of self-reported assessments, general use of active control groups and objective measures, and addressing yoga interventions’ heterogeneity [67].

Concerning qigong, and according to some authors, evidence on QoL improvement has not yet been established [66,68], but according to other authors, qigong seems to be a good strategy for cancer-related symptoms and QoL in cancer survivors [69], at least in women with breast cancer [70].

As described for yoga, statistically significant and clinically meaningful effects of qigong interventions were observed for symptoms of fatigue [71,72], sleep quality [72], and immune function [71] in cancer patients. Additionally, Carlson et al. [73] reported that both qigong and yoga might improve QoL in these patients.

### 4.6. Menopause Symptoms

Shepherd-Banigan et al. reviewed studies on the efficacy of yoga and qigong on the improvement of symptoms of peri- or post-menopausal women, namely, vasomotor symptoms, psychological symptoms, and QoL [74]. The authors did not identify any systematic reviews or RCTs using qigong in this context [74]. Concerning yoga, small to moderate benefits in the reduction of hot flash severity and psychological symptoms were observed, without an impact on QoL [74]. On the contrary, Cramer et al. reported that yoga can reduce psychological, somatic, vasomotor, and urogenital menopausal symptoms [75]. A more recent meta-analysis on menopause symptoms found that yoga significantly improved physical QoL, but its effects on the general, psychological, sexual, and vasomotor symptom QoL scores were not significant [76].

## 5. Mechanisms of Action of Yoga and Qigong

Both yoga and qigong use meditation practices that have been described as counteracting many of the stress responses, presumably by activating the parasympathetic nervous system [41]. It has been proposed that mindfulness meditation leads to increased self-regulation, which comprises attention control, emotion regulation, and self-awareness [77]. This assumption is supported by the observation of changes in the brain with this practice, namely, in the anterior cingulate cortex, which is the area associated with attention; changes in activity and/or structure have been most consistently reported [77].

Yoga training has also been associated with diminished amygdala activation and reduced negative emotion in response to emotional distracter images [14]. This indicates that yoga, due to its neuroplastic effects, may be useful in therapies for certain neurological and psychosocial disorders [14]. Yoga has been suggested to have a neuroprotective effect against the degradation of the brain’s gray matter, which occurs as we age [78]. Moreover, a higher number of regular yoga sessions was associated with larger brain volume in areas involved in body representation, attention, self-relevant processing, visualization, and stress regulation [78].

Yoga has a direct effect on the secretion of sympathetic hormones, such as cortisol and catecholamine, thus improving parasympathetic activity and reducing the metabolic rate [79]. Yoga practice has the capacity to downregulate the hypothalamic–pituitary–adrenal (HPA) axis, which is hyperactivated as a response to abnormal physical or psychological demands [13,26,80]. Yoga attenuates the stress response by reducing the perception of stress [13]. Another mechanism of yoga’s action is through vagal stimulation, leading to improved baroreflex sensitivity and reduced inflammatory cytokines, which, in turn, diminish blood pressure and resting heart rate [26,81,82,83,84]. Furthermore, according to a small but relevant body of research, certain yoga practices may increase melatonin, resulting in beneficial effects on mood, affect, emotion, and mental state [84].

Qigong has also been shown to promote activation of the parasympathetic nervous system by inducing relaxation [41,85,86], to ameliorate immune function, to increase blood levels of endorphins, and to improve baroreflex sensitivity [41,86]. Nevertheless, the effect of qigong on the HPA axis still remains unclear [85].

An important research question is that of the neurophysiological processes that mediate the beneficial effects of qigong [87]. Studies using electroencephalography (EEG) and fMRI have shown changes in brain activity, with most studies reporting increases in theta and alpha activity [87], which indicate a more relaxed state of mind [14]. These alterations were also observed with yoga practice [14], namely, in police trainees after six months of practice [88].

The intentionally controlled breathing in yoga/qigong stimulates the autonomic nervous system, which is widely connected with the internal organs [2]. This helps to build up a state of physiological harmony, which otherwise stays independent of the body’s voluntary control, through the opening of new channels of communication between the internal organs [2].

The relaxation response can translate into an altered gene expression. Enhanced expression of genes in association with energy metabolism, mitochondrial function, insulin secretion and telomere maintenance, and reduced expression of genes in connection with inflammatory response and stress-related pathways have been described after yoga-/qigong-induced relaxation responses [89]. This shows that the relaxation response appears to act at the cellular level to induce the health benefits associated with reducing psychosocial stress [89].

## 6. Limitations

Problematic research issues within the literature on qigong and yoga are usually related to small sample sizes, use of different styles of yoga/qigong, significant variance in practice duration and frequency, short intervention periods [25,86,90,91], and the usage of a non-active control group (e.g., waiting list) [16,25]. With a non-active control group for comparison, we cannot say that the health benefits observed are due to the mind–body intervention with yoga or qigong; they may due to the physical exercise that both practices involve. In fact, Field [92] reported that, although yoga was more effective than a waitlist control condition, it was not always more effective than other forms of exercise.

Almost all studies cited referred to the need for further high-quality randomized trials to provide definitive evidence on the health benefits of yoga/qigong.

## 7. Future Comparative Studies

Considering the limitations referred to above, we envision the realization of high-quality RCTs that compare yoga and qigong in a health context. To achieve this high quality, an active control group should be included in each study in order to distinguish between the effects of the mind–body interventions and the effects resulting from the physical activity and relaxation. To better compare yoga and qigong and due to the existence of different styles of these modalities, which makes comparisons difficult, the yoga protocol should include a slow-movement *asana* sequence plus *pranayama* and meditation.

Several health contexts may be addressed, as it is important to choose a condition for which we can have a good number of participants in order to obtain sufficient sample power to draw valid conclusions. The inclusion of more objective measures, aside from QoL questionnaires and other scales, is also an important aspect to consider. Inflammatory biomarkers are good candidates for this purpose [93].

The duration of the interventions is another critical issue, since most studies used short periods of time. In order to evaluate the efficacy of yoga/qigong, an intervention period of 4 to 6 months with biweekly sessions would be advisable. This is most important if the studies include biomarkers, whose changes are only noticeable after a few months of practice.

## 8. Conclusions

Summing up, yoga and qigong have resulted from thousands of years of experience in using mind–body practices to treat diseases, promote health and longevity, improve fighting skills, and achieve different levels of development of awareness and spirituality.

Yoga and qigong seem to have similar effects; this may be expected, since both are comparable mind–body approaches. There are many similarities between them, and their overall purpose is the same, even though the way in which they achieve it may be slightly different. In general, they have been used for similar health conditions, even though more research has been conducted on yoga in comparison with qigong. Moreover, for yoga, most trials have been conducted on relatively younger healthy participants across India and the United States, while for qigong, most trials have been conducted with relatively older ill people in China and the United States [20].

Participants’ preferences between yoga and qigong apparently differ, but this is probably due to the lower availability of qigong classes and the comparative lack of knowledge about qigong.

Studies comparing yoga and qigong (such as those proposed in the previous section) are warranted in order to assess differences/similarities between the two approaches in the health context. As far as we know, such studies have not been published to date, but are certainly needed.

## Data Availability

Not applicable.

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
