# Peer review of "Yoga and Qigong for Health: Two Sides of the Same Coin?"

_behavsci, 2022, doi:10.3390/bs12070222_

Round 1

Reviewer 1 Report

General Comments

According the authors, It is currently understood that yoga and qigong are two popular systems of self-performed bodily exercises applied to the maintenance of an healthy state of the body and mind. These therapies function as alternative/complementary approaches to manage disease, mainly chronic problems for which there is no effective conventional treatment. So, it is usually referenced as possible and eventual effect on the immune system and inflammation, low back pain, mood disorders, sleep disturbance, cognitive impairment and quality of life, cardiovascular, neurologic, respiratory and metabolic diseases, cancer, menopause symptoms.

However, the authors are faced with the contingency of having to work on a subject that still and only has few effective and clear results. But even so, there is a need to research and begin to address and organize ideas, therapeutic realities and limitations on the subject under study.  In this context, the authors showed some courage to accomplish a bibliographic research, although they could and perhaps even should point out a generic protocol program or perspective of relevant considerations for a future comparative study.

 Suggestion:

-          In this general context, I would like to propose a short chapter, after limitations chapter, concerning an envision of a perceptive protocol program or a general relevant considerations for a future comparative study in order to reach more clarification about the effectiveness of Yoga and Qigong therapy methodology.

Author Response

Reviewer #1 comments

General Comments

According the authors, it is currently understood that yoga and qigong are two popular systems of self-performed bodily exercises applied to the maintenance of an healthy state of the body and mind. These therapies function as alternative/complementary approaches to manage disease, mainly chronic problems for which there is no effective conventional treatment. So, it is usually referenced as possible and eventual effect on the immune system and inflammation, low back pain, mood disorders, sleep disturbance, cognitive impairment and quality of life, cardiovascular, neurologic, respiratory and metabolic diseases, cancer, menopause symptoms.

However, the authors are faced with the contingency of having to work on a subject that still and only has few effective and clear results. But even so, there is a need to research and begin to address and organize ideas, therapeutic realities and limitations on the subject under study.  In this context, the authors showed some courage to accomplish a bibliographic research, although they could and perhaps even should point out a generic protocol program or perspective of relevant considerations for a future comparative study.

 Suggestion:

-          In this general context, I would like to propose a short chapter, after limitations chapter, concerning an envision of a perceptive protocol program or a general relevant considerations for a future comparative study in order to reach more clarification about the effectiveness of Yoga and Qigong therapy methodology.

Authors reply

We are thankful and appreciate the Reviewer comments highlighting the need to do research in the area under study.

According to the reviewer suggestion, we have added to the manuscript a short section with general considerations for a future comparative study with yoga and qigong, entitled “Future comparative studies” (lines 362-379).

Reviewer 2 Report

The article deals with a little explored topic, but which is of scientific interest and it is a significant contribution to the field. The article is accurate, well described and organized, with appropriate and adequate references. 

It may be helpful to find a clearer and more accurate way of describing  points below:

"freedom" line 11

"yoga and qigong were not created by a single individual" line 329.

Referencing PubMed can also be improved a little.

Author Response

Reviewer #2 comments

The article deals with a little explored topic, but which is of scientific interest and it is a significant contribution to the field. The article is accurate, well described and organized, with appropriate and adequate references. 

Authors reply

We are thankful and appreciate the Reviewer comments about the importance and quality of the manuscript.

Reviewer requests

It may be helpful to find a clearer and more accurate way of describing  points below:

Request #1 - "freedom" line 11

Authors reply

According to the reviewer suggestion we have changed the word “freedom”. Instead we used the expression “mind clarity”. We changed it in line 11 and in the Abstract.

Request #2 - "yoga and qigong were not created by a single individual" line 329.

Authors reply

We removed the expression “yoga and qigong were not created by a single individual” in the manuscript.

Request #3 - Referencing PubMed can also be improved a little.

It was not clear to us which improvement the reviewer was referring to.

Reviewer 3 Report

The paper needs some changes in order to be suitable for publication.

INTRODUCTION. First of all, the general idea (arising even from the title) is that the two analyzed practices (Yoga and qigong) are considered as equal. The similiraties are discussed, while the differences are not covered at lenght. For those who are not expert in the field, you must expand on this a bit more. The results section covers together the results pertaining both techniques, but the introduction should contain a justification of your choices. Is literature covering these practices as interchangeably, are you advancing the idea that they should be considered the same?

METHODS. A section clarifying your methods at lenght (databases, seach terms, any inclusion/exclusion criteria) is needed.

GENERAL CONSIDERATIONS. English grammar needs to be revisited. Foe example, line 17...have similar effects what could be expected...please check the whole manuscript for grammar.  Moreover, you should describe a bit more the practical implications of your paper.

Author Response

Reviewer #3 comments

The paper needs some changes in order to be suitable for publication.

Authors reply

We thank the reviewer for the comments which allowed us to improve the manuscript.

Request #1

INTRODUCTION. First of all, the general idea (arising even from the title) is that the two analyzed practices (Yoga and qigong) are considered as equal. The similiraties are discussed, while the differences are not covered at lenght. For those who are not expert in the field, you must expand on this a bit more. The results section covers together the results pertaining both techniques, but the introduction should contain a justification of your choices. Is literature covering these practices as interchangeably, are you advancing the idea that they should be considered the same?

Authors reply

We have considered these comments about the differences between yoga and qigong and added the information in the manuscript. We added, in the Introduction section, a paragraph emphasizing yoga and qigong similarities and differences (lines 52-62), and a justification of our choices (lines 84-92).

We hope this issue is now more well presented in the manuscript.

Request #2

METHODS. A section clarifying your methods at lenght (databases, seach terms, any inclusion/exclusion criteria) is needed.

Authors reply

As suggested by the Reviewer, a section clarifying the methods was added to the manuscript (lines 94-106).

Request #3

GENERAL CONSIDERATIONS. English grammar needs to be revisited. Foe example, line 17...have similar effects what could be expected...please check the whole manuscript for grammar. 

Authors reply

The whole manuscript was checked for grammar; specifically, the expression “have similar effects what could be expected” was replaced “with similar effects which might be expected”.

Request #4

Moreover, you should describe a bit more the practical implications of your paper.

Authors reply

According to the reviewer request we further described the practical implications of the manuscript (lines 84-92).

Reviewer 4 Report

Yoga and qigong for health: two sides of the same coin?

Thank you for the opportunity to review this interesting manuscript. Specific comments below.

15 – believe would be more correct to say ‘a patient’s’ or patients’ 

20 – consider revising ‘short term of the intervention’ for improved readability – such as short duration of intervention effect

34 – remove ‘on’ or revise to ‘place emphasis on’

57 – compared to yoga

83 – compared to only

91 – consider revising to ‘In contrast, a yoga practitioner in India..’

94 – consider revising to ‘ In the USA and Australia, the most common reason to practice yoga is physical fitness, followed by disease management.’

117 -  ‘Of note,’ instead of noteworthy

118 – patients’

124 – remove ‘the’ before psychological distress

128 – remove ‘has’ before may modify

136 – add ‘and’ before respiratory rate

138 – compared to

149 – comma after depression, and remove comma before as

160 – Of note

248 – enrollment not enrolment 

265 – space after period 

315 – seems to act? Revise for clarity

Author Response

Reviewer #4 comments

Thank you for the opportunity to review this interesting manuscript. Specific comments below.

Authors reply

We are thankful and appreciate that the Reviewer found the manuscript interesting.

Request #1  

15 – believe would be more correct to say ‘a patient’s’ or patients’ 

Authors reply

According to the suggestion “patient’s” was replaced with “patients’.

Request #2  

20 – consider revising ‘short term of the intervention’ for improved readability – such as short duration of intervention effect

Authors reply

According to the suggestion revising “short term of the intervention” was replaced with “short duration of intervention effect”.

Request #3

34 – remove ‘on’ or revise to ‘place emphasis on’

Authors reply

According to the suggestion we changed the sentence and used the expression “to place emphasis on”.

Request #4

57 – compared to yoga

Authors reply

We have rephrased to “compared to”.

Request #5

83 – compared to only

Authors reply

We have rephrased to “compared to only”.

Request #6

91 – consider revising to ‘In contrast, a yoga practitioner in India..’

Authors reply

We have revised the sentence as requested.

Request #7

94 – consider revising to ‘ In the USA and Australia, the most common reason to practice yoga is physical fitness, followed by disease management.’

Authors reply

We have revised the sentence as requested.

Request #8

117 -  ‘Of note,’ instead of noteworthy

Authors reply

We replaced “Noteworthy” with “Of note” as suggested.

Request #9

118 – patients’

Authors reply

We have replaced “patients” with “patients’.

Request #10

124 – remove ‘the’ before psychological distress

Authors reply

The word “the” was removed.

Request #11

128 – remove ‘has’ before may modify

Authors reply

The word “has” was removed.

Request #12

136 – add ‘and’ before respiratory rate

Authors reply

The word “and” was added before “respiratory rate”.

Request #13

138 – compared to

Authors reply

We have rephrased to “compared to”.

Request #14

149 – comma after depression, and remove comma before as

Authors reply

We added comma after “depression”, and removed comma before “as”.

Request #15

160 – Of note

Authors reply

We replaced “Noteworthy” with “Of note” as suggested.

Request #16

248 – enrollment not enrolment 

Authors reply

We replaced “enrolment” with “enrollment”.

Request #17

265 – space after period 

Authors reply

We added a space after period.

Request #18

315 – seems to act? Revise for clarity

Authors reply

As suggested by the Reviewer, the sentence was revised for clarity.

Round 2

Reviewer 1 Report

General Comments

According the authors, It is currently understood that yoga and qigong are two popular systems of self-performed bodily exercises applied to the maintenance of an healthy state of the body and mind. These therapies function as alternative/complementary approaches to manage disease, mainly chronic problems for which there is no effective conventional treatment. So, it is usually referenced as possible and eventual effect on the immune system and inflammation, low back pain, mood disorders, sleep disturbance, cognitive impairment and quality of life, cardiovascular, neurologic, respiratory and metabolic diseases, cancer, menopause symptoms.

However, the authors are faced with the contingency of having to work on a subject that still and only has few effective and clear results. But even so, there is a need to research and begin to address and organize ideas, therapeutic realities and limitations on the subject under study.  In this context, the authors showed some courage to accomplish a bibliographic research, although they could and perhaps even should point out a generic protocol program or perspective of relevant considerations for a future comparative study.

Suggestion:

-          In this general context, I would like to propose a short chapter, after limitations chapter, concerning an envision of a perceptive protocol program or some general relevant considerations for a future comparative study in order to reach more clarification about the effectiveness of Yoga and Qigong therapy methodology.

Author Response

Reviewer #1 comments

General Comments

According the authors, it is currently understood that yoga and qigong are two popular systems of self-performed bodily exercises applied to the maintenance of an healthy state of the body and mind. These therapies function as alternative/complementary approaches to manage disease, mainly chronic problems for which there is no effective conventional treatment. So, it is usually referenced as possible and eventual effect on the immune system and inflammation, low back pain, mood disorders, sleep disturbance, cognitive impairment and quality of life, cardiovascular, neurologic, respiratory and metabolic diseases, cancer, menopause symptoms.

However, the authors are faced with the contingency of having to work on a subject that still and only has few effective and clear results. But even so, there is a need to research and begin to address and organize ideas, therapeutic realities and limitations on the subject under study.  In this context, the authors showed some courage to accomplish a bibliographic research, although they could and perhaps even should point out a generic protocol program or perspective of relevant considerations for a future comparative study.

 Suggestion:

-          In this general context, I would like to propose a short chapter, after limitations chapter, concerning an envision of a perceptive protocol program or a general relevant considerations for a future comparative study in order to reach more clarification about the effectiveness of Yoga and Qigong therapy methodology.

Authors reply

We are thankful and appreciate the Reviewer comments highlighting the need to do research in the area under study.

According to the reviewer suggestion, we have already added to the manuscript (after revision Round 1), a short section with general considerations for a future comparative study with yoga and qigong, entitled “Future comparative studies” (lines 366-383). This was the added text:

“Considering the above referred limitations, we envision the realization of high-quality RCTs comparing yoga and qigong in a health context.  To achieve this high quality, an active control group should be included in each study, in order to distinguish between the effects of the mind-body interventions and the effects resulting from the physical activity and relaxation per se. To better compare yoga and qigong, and due to the existence of different styles of these modalities making comparisons difficult, the yoga protocol should include a slow movement asana sequence plus pranayama and meditation.

Several health contexts may be addressed, being important to choose a condition for which we can have a good number of participants, to obtain sufficient sample power to draw valid conclusions. The inclusion of more objective measures, besides QoL questionnaires and other scales, is also an important aspect to consider. Inflammatory biomarkers are good candidates for this purpose (93).

The duration of the interventions is another critical issue, since most studies used short periods of time. In order to evaluate yoga / qigong efficacy, an intervention period of 4 to 6 months with biweekly sessions would be advisable. This is most important if the studies include biomarkers, whose change is only noticeable after a few months of practice.”

We hope that the information added is in line with what was suggested by the reviewer.

Reviewer 3 Report

The paper is now suitable for publication. 

Author Response

Reviewer #3 comments

The paper is now suitable for publication. 

Authors reply

We thank the reviewer for accepting the manuscript.